# Vitamin D Signaling in Gastro-Rheumatology: From Immuno-Modulation to Potential Clinical Applications

**DOI:** 10.3390/ijms22052456

**Published:** 2021-02-28

**Authors:** Cristiano Pagnini, Andrea Picchianti-Diamanti, Vincenzo Bruzzese, Roberto Lorenzetti, Michele Maria Luchetti, Louis Severino Martin Martin, Roberta Pica, Palma Scolieri, Maria Lia Scribano, Costantino Zampaletta, Maria Sole Chimenti, Bruno Lagana

**Affiliations:** 1Department of Gastroenterology and Digestive Endoscopy, S. Giovanni Addolorata Hospital, 00184 Rome, Italy; cpagnini@hsangiovanni.roma.it; 2Department of Clinical and Molecular Medicine, S. Andrea University Hospital, Sapienza University, 00189 Rome, Italy; bruno.lagana@uniroma1.it; 3Department of Internal Medicine, Rheumatology and Gastroenterology, Nuovo Regina Margherita Hospital, 00153 Rome, Italy; vinbruzzese@tiscali.it (V.B.); rlorenzetti2004@yahoo.it (R.L.); palma.scolieri@gmail.com (P.S.); 4Clinica Medica, Dipartimento di Scienze Cliniche e Molecolari, Università Politecnica delle Marche, 60131 Ancona, Italy; m.luchetti@staff.univpm.it; 5Ospedale Regina Apostolorum, U.O. di Medicina Interna e Reumatologia, 00041 Albano Laziale, Italy; severino.martin@gmail.com; 6Unit of Gastroenterology and Digestive Endoscopy, Sandro Pertini Hospital, 00157 Rome, Italy; robertapica5@gmail.com; 7Gastroenterology Unit, AO San Camillo Forlanini, 00152 Rome, Italy; marialiascribano@virgilio.it; 8Division of Gastroenterology, Belcolle Hospital, 01100 Viterbo, Italy; zcosta@libero.it; 9Rheumatology, Allergology and Clinical Immunology, Department of Medicina dei Sistemi, University of Rome Tor Vergata, 00187 Rome, Italy; maria.sole.chimenti@uniroma2.it

**Keywords:** osteoimmunology, inflammatory bowel diseases, spondyloarthritis, microbiota, intestinal mucosal barrier, vitamin D, vitamin D receptor, immunomodulation

## Abstract

In the last decades, the comprehension of the pathophysiology of bone metabolism and its interconnections with multiple homeostatic processes has been consistently expanded. The branch of osteoimmunology specifically investigating the link between bone and immune system has been developed. Among molecular mediators potentially relevant in this field, vitamin D has been recently pointed out, and abnormalities of the vitamin D axis have been described in both in vitro and in vivo models of inflammatory bowel diseases (IBD) and arthritis. Furthermore, vitamin D deficiency has been reported in patients affected by IBD and chronic inflammatory arthritis, thus suggesting the intriguing possibility of impacting the disease activity by the administration vitamin D supplements. In the present review, the complex interwoven link between vitamin D signaling, gut barrier integrity, microbiota composition, and the immune system was examined. Potential clinical application exploiting vitamin D pathway in the context of IBD and arthritis is presented and critically discussed. A more detailed comprehension of the vitamin D effects and interactions at molecular level would allow one to achieve a novel therapeutic approach in gastro-rheumatologic inflammatory diseases through the design of specific trials and the optimization of treatment protocols.

## 1. Introduction

Bone metabolism is a complex and dynamic process that tightly regulates the composition of the skeleton of human body. Besides their structural function, the bones play a fundamental role for hosting, in the bone marrow, hematopoietic cells (HSCs), myeloid and lymphoid progenitors, and mature cell of the immune system. Those cells share the same milieu of the cells regulating the bone metabolism (e.g., osteoblasts, osteoclasts, and osteocytes) and are closely connected by reciprocal interactions mediated by multiple molecular mediators, such as cytokines, chemokines, transcription factors, and signaling molecules [1]. The potential relation between osteogenesis and immune system has been highlighted since the 1970s in studies regarding periodontitis [2]. In 2000, the term “osteoimmunology” was coined to define the complex interwoven link between these two systems, particularly evident in T-cells mediated regulation of osteoclastogenesis observed in autoimmune arthritis [3]. Multiple molecular mediators have shown a potential role in the osteoimmune network [4]. Osteoblasts progenitors produce stem cell factors and CXC-chemokine ligand 12 (CXCL12) that are crucial for HSC maintenance and differentiation, and mature osteoblasts produce interleukin-7 (IL-7) that has an important role in the regulation of the lymphoid lineage. Osteoclasts produce proteolytic enzymes, such as matrix-metallopeptidase 9 and cathepsin K, that contribute to the HSC mobilization. Moreover, the bone reabsorption process is essential for the bone marrow cavity formation as well as for the increase in calcium level and the release of some factors, e.g., transforming growth factor (TGF)-β), that have a role in HSC regulation. Osteocytes regulate lymphoid and myeloid differentiation through the production of sclerostin and granulocyte colony-stimulating factor (G-CSF). Conversely, the activated immune system and the aberrant inflammation may affect osteosynthesis through the production of IL-17 by Th17 cells and the induction of the receptor activator of nuclear factor-κB ligand (RANKL) further amplified by the pro-inflammatory cytokines IL-1, IL-6, and tumor necrosis factor (TNF), which promote osteoclastogenesis. In line with this vision, the bone alterations observed in several immune diseases are no longer considered related merely to malnutrition or steroids use, and the common osteoimmune molecular pathway has been proposed as a novel potential target for therapeutic strategies.

### 1.1. Vitamin D and VDR Expression and Biologic Effects

One of the molecular mediators of the osteoimmunity regulation that has been intensely studied in the last years is the vitamin D, a fat soluble secosteroid hormone that is present in two main forms: vitamin D2 (ergocalciferol), from mushrooms and, in minor quantity, from vegetables, and vitamin D3 (colecalciferol), from animal origin. Besides alimentary sources that are substantially scarce, vitamin D3 is endogenously synthetized in the skin for the transformation by the UV light of the cholesterol precursor 7-dehydrocholesterol in pre-vitamin D3 and then in vitamin D3 [5]. Once in the circulation, vitamin D3 and D2 are converted in the liver by the enzyme 25-hydroxylase (CYP2R1) in 25 hydroxyvitamin D (25(OH)D) and then in the kidney by the enzyme 1-α-hydroxylase (CYP27B1) into its active form, 1,25-dihydroxyvitamin D (1,25(OH)_2_D or calcitriol). Additionally, some peripheral tissues (such as intestinal epithelial cells) and immune cells also express CYP27B1, therefore a local para- and autocrine function may play a role in vitamin D signaling (Figure 1).

In the circulation, vitamin D is largely bound to vitamin D binding proteins (DBP) and albumin, whereas the minor unbound fraction is most probably the only one capable of entering the cells’ cytoplasm and exerting the biological effects [6]. The binding of 1,25(OH)_2_D to VDR in the cytoplasm of the cell determine the heterodimerization with the retinoid X receptor (RXR) and other cofactors, and the translocation of the complex to the nucleus, where it binds to vitamin D response elements (VDREs), either promoting or suppressing gene transcription [7,8]. VDR is a single aminoacidic chain polypeptide that belongs to the nuclear receptors superfamily. It is widely distributed but with relevant differences among tissues [9]. Indeed, it is highly expressed in tissue involved in bone metabolism such as bones, intestine, cartilage, kidney, and parathyroid glands, where it was firstly identified. It is also present in extra-skeletal tissue such as immune system, heart, adipose tissue, respiratory tract, and dermal fibroblasts/keratinocytes of skin [10,11], whereas it is poorly expressed in tissues such as muscle, liver, and particularly the central nervous system [12,13,14].

The biologic actions are mediated not only by genomic but also by epigenomic pathways by means of inducing the expression of chromatin modifier enzymes and by direct interaction with chromatin proteins that in turn regulate epigenetic post-translational events [15]. Analysis of the genome-wide binding sites of the VDR complex has been recently performed by the chromatin immunoprecipitation combined with sequencing (ChIP-seq) method. These works showed an average more than 10,000 VDR binding loci per cell type, most of them being specific for a single cell line and variably contributing to the modulation of hundreds of vitamin D target genes in VDR expressing tissues. In an ex vivo study with short term cultured biopsies from healthy colonic tissue treated with 1,25(OH)_2_D, genome wide transcriptional profiling identified 465 upregulated and 417 downregulated genes [16]. In a study of intestinal epithelial organoids from normal colonic mucosa, ChIP-seq identified, after 1,25(OH)_2_D stimulation for 2 h, an association for 182 loci that were linked with genes differentially expressed after 1,25(OH)_2_D treatment. Among those, the most significant genes were *CYP24A1* (responsible for 1,25(OH)_2_D degradation), *TRPV6* (an intestinal calcium channel), and *CD14* (a coreceptor in TLR4 signaling) [17]. In immune cells, a recent study on vitamin D-stimulated monocytes, with ChIP-seq and formaldehyde-assisted identification of regulatory elements followed by sequencing (FAIRE-seq) methods (which determine genome-wide chromatin accessibility), highlighted a short list of 15 genes as major targets of vitamin D in human immune system. These genes were categorized in three groups according to their basal activity and inducibility: group 1 (*CAMP, CD14, FN1, TREM1*) includes proteins related to acute response to infection, with low basal expression but high inducibility, group 2 (*LILRB4, LRRC25, MAPK13, SEMA6B, THBD, THEMIS2*) refers to proteins involved in general response to infection, with intermediate responsiveness to transcriptome and epigenome level, and group 3 (*ACVRL1, CD93, CEBPB, NINJ1, SRGN*) is represented by proteins related to autoimmunity, with high basal activity and low inducibility genes [18].

The biologic action of vitamin D/VDR signaling is pleiotropic; besides its early clarified role in the bone metabolism, many other homeostatic functions are directly and indirectly influenced. Different pathologic conditions have been found to be related to alteration of the vitamin D pathways, among which are immuno-mediated pathologies.

### 1.2. Gut–Joint Axis

Inflammatory bowel diseases (IBDs) and spondyloarthritis (SpA) are systemic diseases with unknown etiology and immunomediated pathogenesis. IBDs include two major forms, ulcerative colitis (UC) and Crohn’s disease (CD), that are mainly characterized by chronic intestinal inflammation with typical intermittent/recurrent clinical symptoms and different localization and histologic involvement. SpA can be classified in axial and peripheral SpA, with the former including ankylosing spondylitis (AS) with and without (non-radiographic AS) the typical radiologic pattern, and the latter psoriatic arthritis (PsA) as well as reactive and IBD-associated arthritis. Despite the striking clinical peculiarities, these conditions share several clinical and pathogenetic features. Cohort studies report a potential incidence of rheumatic involvement in up to 40% of IBD patients; on the other hand, 15–20% of SpA patients present concomitant IBD, but this rate can consistently raise up to 50% considering sub-clinical gut inflammation [19,20,21]. Both diseases are complex multifactorial disorders in which genetic background and environmental factors variably contribute to the onset and the maintenance of a deregulated immune response that in turn promotes the clinical disease phenotype. Genome-wide association studies (GWAS) have identified almost 120 genetic loci linked to AS and nearly 240 to IBD with a rate of shared risk alleles between those two diseases higher than in other autoimmune disorders [22,23]. In particular, many genes related to type 3 immunity and epithelial barrier integrity are shared between AS and CD [24]. The type 3 immunity is a polarized arm of the immune system directed toward extracellular fungi and bacteria and characterized by the production of IL-17 and IL-22 by Th17 and other cell types. IL-17 is likely to play a dichotomous role in gastro-rheumatology. IL-23 independent production by the innate cells of the intestinal mucosa appears to have a protective role by the stimulation of tight junction interactions and anti-bacterial peptides production, while IL-23 dependent production by Th17 cells promotes deregulated chronic inflammation through the production of other pro-inflammatory cytokines such as TNF, IL-1, and IL-6, activated T-cells recruitment, and radical oxygen species (ROS) production [25,26]. Th17 cells are involved both in SpA and IBD pathogenesis, with increased concentration of IL-23 and IL-17 in small bowel of CD patients, and IL-17 elevated concentration in serum and in the synovium of SpA patients [27,28]. Biologic drugs targeting IL-17 (e.g., secukinumab, ixekizumab) showed consistent efficacy and are currently approved for treatment in peripheral and axial SpA, while they have no effect or can even exacerbate IBD, further confirming the dual effect of these cytokines in systemic and intestinal mucosal inflammation [29,30]. The intestinal barrier is a complex system that separates and regulates the bi-univocal interaction between the intestinal luminal content (mainly constituted by the microbiota) and the submucosal compartment, which is the entrance door to the human organism and the frontline of the immune system. In IBD and SpA, alteration of the intestinal barrier function and increased permeability have been identified as early pathogenetic events for the onset and the development of the chronic deregulated inflammatory process. Besides the sole restraining role, the intestinal barrier is able to sense and modulate the luminal content, thus influencing the composition of the complex ecosystem falling under the term of microbiota. In the last decades, intense research has highlighted the paramount influence of the microbiota in maintaining the physiologic state in the organism and the possibility that specific perturbations could drive to different pathologic conditions through multiple interactions [31]. Notwithstanding specific molecular pathways that may differ between the two groups of diseases, IBD and SpA are therefore indisputably interconnected from a pathogenetic point of view. The increasing knowledge of the so called “gut–joint axis” is leading to a modern vision of intestinal and rheumatic inflammatory chronic diseases in which, in order to better encompass similarities and differences, the research and the clinical practice need to be interdisciplinarily linked and bi-univocal, identifying the novel field of “gastro-rheumatology”.

In the present paper, we intended to review the complex interwoven relationship among vitamin D/VDR pathway, immune system, and gut microbiota and to critically analyze the current knowledge and the potential clinical applications in gastro-rheumatologic inflammatory diseases.

## 2. Literature Search

Online literature searches were conducted to identify English language publications reporting preclinical studies, clinical trials, and real-world data evaluating expression, function, and biology of vitamin D and vitamin D receptor, in particular focusing on their interaction with the immune system and the role in inflammatory bowel diseases and arthritis. We performed a search of the PubMed database including several interrelated queries: “vitamin D”, “vitamin D receptor”, “inflammatory bowel diseases”, “ulcerative colitis”, “Crohn’s disease”, “microbiota”, “spondyloarthritis”, and “arthritis”. The search is updated on 15 January 2021. Given the narrative nature of the review, the articles retrieved were chosen according to their relevance, as judged by the authors.

## 3. Vitamin D and Intestinal Permeability

Vitamin D is a dietary nutrient with demonstrated anti-inflammatory and immunomodulating functions. Recent data suggest that the intestinal mucosal barrier is a possible *trait d’union* between vitamin D, immune system, and gut microbiota. The intestinal mucosa is both an absorption site that allows entry of food-derived metabolites and a physical barrier that blocks pathogens translocation, thus protecting against infection with enteropathogenic microorganisms and intestinal inflammation. The intestinal epithelium is composed by enterocytes and specialized epithelial cells, such as Goblet and Paneth cells. Goblet cells produce mucus that forms a layer between the epithelium and the luminal contents, whereas Paneth cells release antibacterial molecules (e.g., α- and β-defensins, cathelicidin) [32]. Sub-mucosal cells of the innate immunity, such as macrophages and dendritic cells (DCs), clear microorganisms and luminal particles that penetrate the first line of defense of the epithelial/mucus layer, thus containing the immune inflammatory reaction. The presence of different structures between adjacent epithelial cells, such as tight junctions (occludin, proteins of the zonula occludens, and claudins), adherens junctions (E-cadherin, catenins, nectin), and desmosomes, is also essential in maintaining the resistance of the intestinal mucosa [33].

In view of the above, it is not surprising that dysregulation in these components, such as defective expression of defensins, upregulation of claudin-2, or increased apoptosis of epithelial cells, can contribute to the disruption of the mucosal barrier, as reported in IBD and SpA patients [34,35,36,37,38]. In particular, it has been hypothesized that, in SpA patients, the increased intestinal permeability, probably induced by genetic factors (HLA-B27), could induce a disruption of the basal membrane, hyperplasia of goblet cells, and activation of Paneth cells producing high levels of anti-microbial peptides (AMPs) and IL-23, leading to exaggerated antigenic stimulation and activation of effector T-cells of the intestinal mucosa [27,39,40,41,42,43].

Vitamin D/VDR signaling can modulate the number and the functionality of tight junction proteins in both in vitro and in vivo studies on transgenic mice. VDR knockout and vitamin D-deficient mice displayed epithelial barrier dysfunction with hyperfunction of claudin-2, decreased transepithelial resistance, and increased susceptibility to invasive bacteria colonization and colitis [44,45,46,47]. Conversely, transgenic mice overexpressing VDR in the gut epithelium have resistance to colitis with decreased mucosal inflammation and apoptosis of epithelial cells [48]. In addition, vitamin D supplementation has been shown to ameliorate the clinical symptoms and the histologic findings in Dextran sulphate sodium (DSS) treated mice by preserving the expression of E-cadherin, claudin, and zonula occludens in Caco-2 cells [49].

## 4. Vitamin D, Mucosal Immunity, and Immunomodulation

As above reported, vitamin D plays an important role in preserving intestinal barrier functionality, however, the complex relationship between vitamin D/VDR, gut microbiota, and the immune system is related to several other mechanisms still only partially understood. At birth, the mucosal immune system is not completely developed because many of the gut lymphoid structures need the exposure to microorganisms to arise. This is showed by in vivo studies in germ free mice, which have small Payer’s patches and lack isolated lymphoid follicles in their bowel as well as decreased serum levels of Immunoglobulin (Ig)M natural antibodies and IgA secreting cells in the spleen and the gut. Moreover, when these mice are exposed to conventional housing and food, bacterial products (such as LPS) are sensed by enterocytes through TLRs, and this immunologic phenotype is reverted [50]. At birth, colonization is of pivotal importance in promoting and shaping the development of the secondary lymphoid organs and setting up thresholds of reactivity for both innate and acquired immune responses. Bacterial-derived products such as the short-chain fatty acids (SCFAs) and the lipid mediator prostaglandin E2 (PGE2) can also influence the activity of epithelial and inflammatory cells. Recognition of the SCFAs by innate immune cells, indeed, is important in modulating inflammation in response to intestinal and articular damage [51]; moreover, commensals bacteria by inducing the release of PGE2 can down-modulate the activation of tissue damaging neutrophils [52].

The cathelicidin AMP (cAMP) expressed by innate cells such as DCs, natural killer (NK) cells, and gut specialized epithelial cells such as the Paneth cells, provides antibacterial activity against both Gram positive and Gram negative bacteria [53]. Defensins are a family of microbicidal and cytotoxic peptides made by neutrophils; in particular, human beta-defensin-2 (encoded by the defensin-beta-4 gene) is released by Paneth and other epithelial cells and is involved in the innate immune response in the gut [54]. The active form of vitamin D can stimulate cAMP and beta-defensin 2 expression in monocytes and macrophages through the activation of TLR2/1, leading to enhanced phagolysosomal and anti-microbial function [55,56]. Conversely, a vitamin D deficient diet or a lack of VDR can lead to impaired anti-bacterial activities of epithelial cells and consequently increased inflammatory responses [57,58]. Gubatan et al. have recently shown a direct correlation between serum vitamin D and colonic cathelicidin in a cohort of UC patients, moreover, vitamin D treatment of colon epithelial cells showed to induce cathelicidin and IL-10 secretion [59].

Autophagy is another essential system involved in the innate immunity responses to encapsulate and destroy bacteria, viruses, and other dangerous substances. Deficits in the autophagy mechanism can alter Paneth cell function and contribute to gut inflammation and dysbiosis. Low levels of intestinal epithelial VDR correlate with a reduced expression of the autophagy gene ATG16L1 and impaired antimicrobial function of Paneth cells [47,60] and can induce gut dysbiosis in patients with IBD [34].

The gut is involved in B-cell development and is a main site for the generation of inducible regulatory T-cells (iTregs) and Th17 cells, both of which play critical roles in the pathogenesis of autoimmune disorders. Segmented filamentous bacteria colonization of the distal small intestine can raise to the formation of resident lamina propria DCs that can release IL6 and IL22 and stimulate the T-cell compartment, in particular the Th17–Treg cell axis in the newborn gut [51,61]. DCs can access luminal antigens and promote B cell differentiation and class switch recombination into IgA producing cells; moreover, plasmablasts move to the intestinal lamina propria where they differentiate into plasma cells [62].

Different in vitro studies and experimental models have shown that vitamin D/VDR signal can regulate the adaptive immunity system by inhibiting Th1, Th17 cells, and DCs differentiation and promoting Treg cells with a consequent reduced production of inflammatory cytokines such as IL-17A, TNF-alpha, IL-6, and interferon-gamma (IFN-γ) [63,64,65].

In particular, the VDR signal can impair the functionality of Th17 cells by inhibiting the binding of transcription factors such as the nuclear factor of activated T-cells and the runt-related transcription factor 1 to Foxp3 [66,67]. Furthermore, vitamin D can induce an expansion of Treg cells and upregulate the expression of CTLA-4 and Foxp3 regulatory markers [68]. On the other hand, T-cells from VDR KO mice produce higher levels of IFN-γ and IL-17 than their wild type counterparts [69,70,71].

Despite these interesting data, studies on autoimmune diseases are scarce and contrasting. In particular, it has been reported that vitamin D supplementation is able to increase Treg function in diabetes mellitus patients, whereas insignificant effects were shown in Rheumatoid Arthritis (RA) and IBD patients [72,73,74,75].

## 5. Vitamin D and Gut Microbiota Homeostasis

The gut microbiota is a complex ecosystem of archaea, bacteria, fungi, and viruses that is essential for digestion of complex carbohydrates as well as absorption and supply of vitamins, but it exerts also immunomodulatory, metabolic, and anti-infective functions. Any imbalance in the gut microbiota resulting in a loss or overgrowth of a species and/or reduction in microbial diversity is defined as dysbiosis. In the last two decades, dysbiosis of the gut microbiota has been described in different pathologies such as depression, IBD, RA, and SpA [76,77,78].

The impact of diet and nutrients on the gut microbiota is suggested by the differences in its composition/variety between geographically and life-style distant populations [79]. It is known, indeed, that a western diet rich in animal proteins, simple sugars, and saturated fats is characterized by a reduction in the variety of microbiomes and is associated with the *Bacteroides* enterotype, whereas a diet habit rich in fruits and vegetables leads to a prevalence of *Prevotella* [80]. Dietary intervention can also impact the gut microbiota composition and richness. Foods rich in fibers, such as those present in the Mediterranean diet (MD), indeed, are degraded by Firmicutes and Bacteroidetes into SCFA, such as butyrate [81,82], which can have a protective role on the gut barrier by reducing its permeability. We have recently found that RA patients with high adherence to MD have a lower disease activity joined to a healthier gut microbiota composition with a significant decrease in *Lactobacillaceae* and an almost complete absence of *Prevotella copri* in comparison with the low/moderate adherence patients [83].

It has been also shown that vitamin D can also influence the composition of the gut microbiome in animal models [84,85]. VDR KO mice with defective autophagy have consequent gut dysbiosis with depletion of *Lactobacillus* and *Bacteroides.* Moreover, administration of butyrate can increase intestinal VDR expression and suppress inflammation in an experimental colitis model [56].

Results on human studies have been recently summarized in a systematic review by Waterhouse et al. [86]. Most of the fourteen analyzed studies evaluated both microbiota diversity and composition and reported significant association between vitamin D and specific changes in gut microbiota. However, there was scarce consistency in the taxa affected and the direction of effect. Indeed, results are hard to compare due to several variables, in particular the heterogeneity in study designs (e.g., cross-sectional vs. prospective, randomized trials vs. observational study), the differences in the assessment of vitamin D (e.g., self-reported dietary, nutritional supplement vitamin D i6ntake, serum 25(OH)D administration), and in the population setting (e.g., healthy people, IBD, cystic fibrosis, multiple sclerosis, infants, pregnant women). Moreover, most of the studies were conducted on very limited samples, and only some of them adjusted for confounding factors such as body mass index, smoking, physical activity, comorbidity, and therapy. Three studies evaluated the effect of vitamin D on the gut microbiota in UC and CD patients. Administration of vitamin D demonstrated a positive effect in modulating the intestinal bacterial composition in both CD and UC patients, leading to a reduced intestinal inflammation in patients with active UC, with a concomitant increase in *Enterobacteriaceae* without changes in microbial diversity [87,88,89]. An additional study published in 2020 was in contrast with these results. In fact, the authors found that reduced levels of vitamin D observed in winter/spring were associated with more balanced microbiome composition both in UC and CD. In particular, they identified lower level of *Escherichia/Shigella* in stool of UC patients and increased level of Bacteroidetes in the stool of CD patients accompanied by lower proportion of *Clostridium spp*. and higher proportion of Firmicutes in the mucosa [90]. Another study that evaluated samples of the intestinal mucosa found a decrease in gammaproteobacteria and increased Bacteroidetes in the microbiome of the upper gastrointestinal tract of patients receiving vitamin D supplementation without significant effects on terminal ileum, ascending colon, sigmoid colon, and stools [91]. Of note, the only GWAS study demonstrated that the VDR gene variation correlated with beta diversity in both humans and mice [92].

Assuming that microbiota and vitamin D have a bidirectional and possible feedback interaction, few studies have evaluated the role of bacteria in modulating vitamin D levels. In fact, it is known that both commensal and pathogenic bacteria can regulate VDR expression and location in mice [93]; some bacteria have enzymes involved in the hydroxylation of steroids and can process and activate vitamin D [94]. Butyrate produced by some gut microorganisms such as Firmicutes and Bacteroidetes can increase VDR expression in the epithelial cells of mice models [47]. In addition, the microbiota can influence vitamin D metabolism through the fibroblast growth factor (FGF)-23 (the protein that regulates the 1,25(OH)2D3 hydroxylating enzyme, CYP27B1). Germ free mice, indeed, have low levels of vitamin D and high FGF-23, and their colonization with bacteria leads to normalization of vitamin D levels and reduced FGF-23 [95].

## 6. Potential Clinical Implications

Bearing in mind the biological actions of vitamin D above described, the possibility of its therapeutic utilization in gastro-rheumatic inflammatory conditions appears intriguing (Figure 2).

In particular, potential association with IBD/SpA onset and severity and vitamin D has been described, and potential therapeutic application in such patients has been explored. To date, studies evaluating vitamin D levels in patients with coexisting IBD and SpA are not available, and only one study included together patients with IBD and SpA. In a multi-center cross-sectional study including 200 patients in clinical remission (136 SpA and 64 IBD), vitamin D deficiency was found to be associated only with biologic therapy at the multivariate analysis both in IBD and SpA patients [96]. Clinical studies have investigated, in IBD and SpA, potential association with vitamin D levels and disease occurrence and/or severity, frequency rate of polymorphisms of genes involving the vitamin D pattern and, finally, have explored the potential impact of vitamin D supplement as a treatment.

### 6.1. Vitamin D Levels in IBD and SpA

In a recent umbrella review including 183 estimates in 53 meta-analyses of 71 environmental factors potentially linked to IBD, vitamin D deficiency has been found among the nine risk factors of increased susceptibility and high vitamin D level among the seven protective factors, with moderate to high strength of epidemiologic evidence [97]. In fact, a meta-analysis including 14 studies with 1891 participants (938 IBD cases and 953 controls) showed that IBD patients had a 64% higher odds of vitamin D deficiency compared to controls and particularly to UC patients [98]. In a very recent meta-analysis including 27 studies with a total of 8316 IBD patients (3115 UC, 5201 CD), low 25(OH)D level was associated with higher risk of disease activity, mucosal inflammation, low quality of life (QOL) scores, and future clinical relapse [99]. Moreover, CD and UC patients with low vitamin D level had higher risk of surgery [100,101].

Among SpA group, axial SpA (axSpA) is associated with reduced serum 25(OH)D levels when compared with healthy controls, and vitamin D deficiency is observed in almost 20% of axSpA patients and correlates with increased disease activity and functional impairment [102]. In fact, Zhao et al. evaluated vitamin D deficiency in 235 axSpA and demonstrated increasing tertiles of Back Ankylosing Spondylitis Disease Activity Index (BASDAI) and VAS pain in patients with vitamin D deficiency. In particular, the highest tertile of BASDAI had a three-fold increased likelihood of vitamin D deficiency compared to those in the lowest BASDAI subgroup. A greater than four-fold increase in likelihood of vitamin D deficiency was observed in patients of the highest VAS pain tertile compared to the VAS pain reference group [103]. Conversely, vitamin D levels were evaluated in newly diagnosed axSpA patients, and no differences were observed between patients and healthy controls. In a cohort of 113 newly diagnosed axSpA patients, vitamin D levels were evaluated, and no association between vitamin D and pain or disease activity was observed. However, female patients had a higher risk of vitamin D insufficiency than male axSpA patients, making gender a predictive variable for low vitamin D levels in these patients [104]. A larger study population, such as data emerging from the ASAS-COMOSPA study initiative, an international cross-sectional study of patients with SpA, demonstrated vitamin D deficiency in 51.2% of the 1030 patients (not receiving any supplementation). Vitamin D deficiency was independently associated with the presence of radiographic sacroileitis supporting the hypothesis that vitamin D deficiency is common in SpA worldwide and is associated with more severe forms of SpA [105].

Four common polymorphisms identified for VDR gene, which is located on chromosome 12, have been investigated for a potential association with SpA and IBD: *ApaI* (rs7975232 A > C), *BsmI* (rs1544410 C > T), *TaqI* (rs731236 T > C), and *FokI* (rs2228570 C > T). *ApaI* a/a was demonstrated to be a protective factor for PsA and for women with PsA, and *ApaI* A/a was a protective factor for PsA patients who were HLA-B27 positive. In this context, the *ApaI* a allele correlated with higher levels of vitamin D in the serum, and *FokI* polymorphism was associated with lower bone mineral density [106,107,108]. In a recent meta-analysis pooling data from nine studies, Xue et al. found a significant increase in CD risk for *TaqI* polymorphism in Europeans and in UC risk for *FokI* polymorphism in Asians [109].

Both conventional disease modifying anti-rheumatic drugs (csDMARDs) and biologic DMARDs (including hydroxychloroquine) have been associated with variations of vitamin D levels in inflammatory conditions [110]. Patients treated with anti-TNF had lower BASDAI scores, but vitamin D levels were similar to healthy controls [111]. A Turkish study enrolled 62 AS patients, and VDR levels were measured; a significant difference was found between patients taking non-steroidal anti-inflammatory drugs (NSAIDs) versus anti-TNF therapy, suggesting that anti-TNF may suppress not only disease activity but also serum VDR. In addition, serum VDR levels in AS patients treated with NSAIDs were elevated compared to the control group, and were significantly increased in the AS group with peripheral joint involvement and enthesitis. The authors suggested that serum VDR level may be used as a marker of disease activity in AS and may have an immunomodulatory function in AS clinical patterns [112].

### 6.2. Vitamin D Supplementation in IBD and SpA

A recent meta-analysis explored current evidence from available studies of therapeutic efficacy of vitamin D administration in IBD patients, evaluating 18 studies with a total of 908 IBD patients. Vitamin D supplement improved the 25(OH)D levels more significantly than the control group with a more consistent effect in high dose vs. low dose administration group. Consistent reduction of relapse rate in vitamin D-treated vs. untreated patients has been observed according to seven trials [113].

Controlled trials investigating the potential therapeutic use of vitamin D in SpA are still lacking, and further studies are required [114], as confirmed by Nguyen et al. in a systematic review and meta-analysis performed last year on the efficacy of vitamin supplementation in inflammatory rheumatic disorders. Only eight studies were included in this review, all on RA whereas no studies on SpA or PsA were selected [115]. The consistent dishomogeneity in basal population, vitamin D treatment protocols and doses, time-point of observation, and the small numerosity of the single studies suggest extreme caution in the interpretation of the results. Moreover, the substantial seasonal variation in serum 25(OH)D levels should be accommodated for, and several studies in the past have neglected this.

Despite the potential relevant role of vitamin D pathway in the pathogenesis and in the maintenance of the chronic gastro-rheumatologic inflammatory diseases, clinical data are weaker than expected, and vitamin D supplements are not considered at present to be a proved efficacious therapy in such conditions. Considering the association between vitamin D deficiency and IBD/SpA occurrence, many potential confounding factors need to be taken into account. First, IBD and SpA are the results of complex and multiple interactions between environmental and genetic factors, thus it is hard to extrapolate the real impact of a single component. Moreover, the vitamin D deficiency is a “pandemic” issue that can involve up to 90% of the population in northern countries where sunlight exposure is lower [116]; in this setting, high association rate in specific subgroups such as IBD or SpA patients may be not necessarily rated to a specific predisposition but only to the wide-spreading of the phenomenon. As for clinical studies in other settings, clinical trials investigating the potential role of vitamin D supplement in IBD/SpA probably suffer basic flaws responsible for the disappointing results [117]. First of all, as vitamin D is a dietary supplement and not a drug, linear dose response is not to be expected. Considering the aforementioned diffuse vitamin D deficiency status in the general population and particularly in IBD/SpA patients, stable serum levels correction needs to be achieved in patients prior to obtaining a significant clinical benefit. Considering this issue, the same “adequate” serum levels of the vitamin D probably need a clearer definition. In fact, at present, most of the studies evaluating vitamin D levels refer, in accordance with the Endocrine Society Guidelines, to “deficiency” for a level of 25(OH)D below 20 ng/mL and to “insufficiency” for a level of 21–29 ng/mL [118]. Notably, these values are set considering the effect of vitamin D in bone metabolism, and it is not clear whether they are fully shiftable to its immunomodulatory activity. In this regard, considering many confounding variables potentially influencing vitamin D level (e.g., diet, obesity, comorbidities, active inflammation), further studies are needed to specifically assess optimal serum level of vitamin D and its potential relation with specific clinical outcomes in IBD and SpA patients, ideally leading in the future to a definitive vitamin D level-based treat-to-target approach. Moreover, in order to fully evaluate the effect of vitamin D in chronic inflammatory diseases, besides blood level, VDR expression in target organs is likely to be taken into account. In fact, some of the most common VDR polymorphisms, such as TaqI and FokI, are likely to affect VDR expression and functionality. Since omozygotic altered genotypes could be present in almost 15–20% of IBD and SpA patients [106,109], possible altered response to vitamin D supplements in this subset of patients needs to be considered. In addition, VDR expression can be induced or inhibited by numerous endogen and exogen stimuli, and in particular inflammatory mediators may downregulate expression of the receptor and therefore prevent vitamin D local effects, even in the presence of an adequate serum amount of the vitamin. Finally, as for many receptors, the interaction with the ligand may downregulate its VDR expression [119], thus reducing potential biologic efficacy of the vitamin for high dose and/or prolonged administration. Since the vitamin D/VDR interaction is fundamental for the biological effects, future studies should more intensely investigate optimal vitamin/receptor balance in order to achieve clinical efficacy.

## 7. Conclusions

In conclusion, the vitamin D/VDR pathway appears as a rational and fascinating field of research with potential development in the field of gastro-rheumatology. At present, besides the potential mechanisms by which vitamin D could interfere with intestinal and articular inflammation, which are being intensely investigated and partly clarified, clinical evidence for a potential therapeutic application is still lacking. With the growing comprehension of the pathogenesis and of the reciprocal interaction of IBD and SpA, it is desirable that novel potential therapeutic strategies will be explored and implemented, and, among those, the vitamin D/VDR pathway appears promising and deserving of further research.

## Figures and Tables

**Figure 1 ijms-22-02456-f001:**
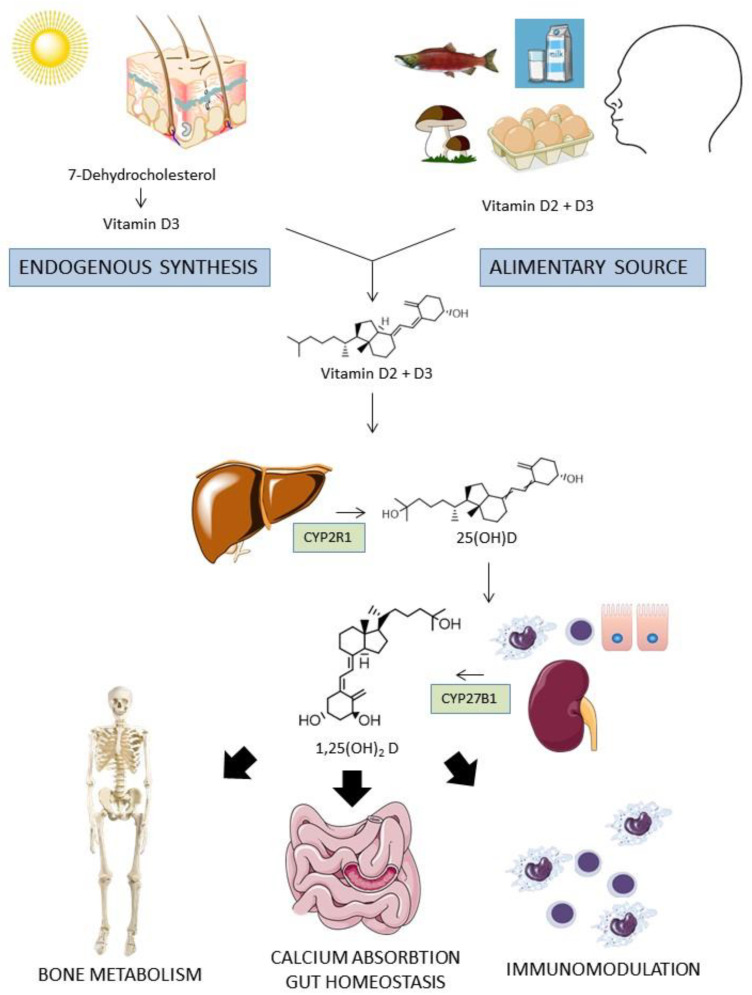
Metabolism of vitamin D. Vitamin D from dietary sources or endogenously synthetized reaches blood circulation. After a double phosphorylation process in the liver and in the kidney (or in some epithelial and immune cells), the active form 1,25 (OH)_2_ D reached many target organs to exert pleiotropic actions, including calcium absorption regulation, bone metabolism, intestinal mucosal homeostasis regulation, and immunomodulation.

**Figure 2 ijms-22-02456-f002:**
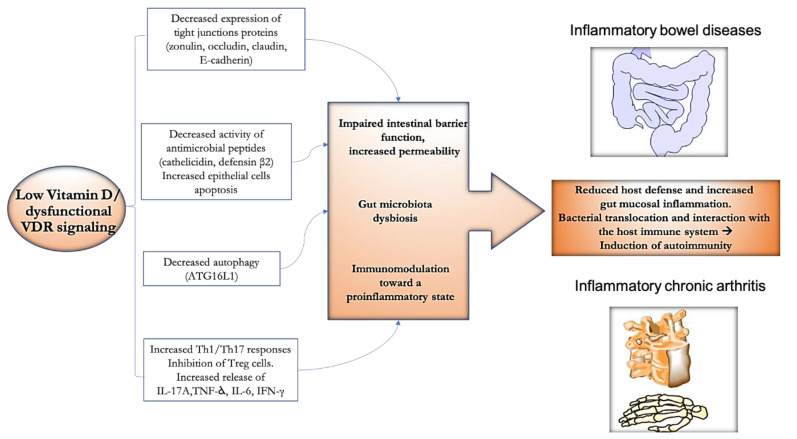
Potential effect of vitamin D towards the gut–joint inflammatory axis. Abnormalities in vitamin D axis could contribute at multiple levels to the dynamic vicious circle, potentially leading to the onset/maintenance of inflammatory bowel diseases (IBD) and spondyloarthritis (SpA). Reduced function of anti-bacterial molecules, decreased expression of tight junctions proteins and autophagy, can lead to impaired intestinal barrier function and increased permeability with a downstream effect on the gut microbiota composition. The unbalance of the Th17/Treg axis toward the former can lead to increased release of proinflammatory cytokines. Reduction of host defense, increased mucosal inflammation, translocation of bacterial products across the mucosal barrier, and interactions with the host immune system would result in the induction of chronic inflammation and autoimmunity.

## Data Availability

Data sharing not applicable.

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
