# Peer review of "Vitamin D Signaling in Gastro-Rheumatology: From Immuno-Modulation to Potential Clinical Applications"

_ijms, 2021, doi:10.3390/ijms22052456_

Round 1

Reviewer 1 Report

Dear Authors,

This is a narrative review in my opinion.

Please respect the format of an article: material and methods, results, discussion, conclusion. 

Please tell us in which database you search and which word you used

Please tell us how many articles was published after search 

Author Response

This is a narrative review in my opinion.

Please respect the format of an article: material and methods, results, discussion, conclusion.

Please tell us in which database you search and which word you used

Please tell us how many articles was published after search

We thank the reviewer for the positive comments on our work.

This is indeed a narrative review in our intention. As requested, we added a material and methods section explaining the database and the terms used, and the period of the literature research. However, considering that it wasn’t a systematic review we didn’t perform a structured a priori selection of the studies, but simply checked their quality and suitability for the purposes of our manuscript. For the same reason, we did unify results and discussion section in a multi-paragraph section after Material and Methods, where we presented and discussed available molecular and clinical data on vitamin D/VDR signaling in IBD and SpA. 

Reviewer 2 Report

Major comments:

  1. Some immune cells also express CYP27B1 and produce 1,25D in a para- and autocrine fashion. this should be discussed and illustrated in Fig. 1.
  2. Please carefully check the relevance of the chosen references. For example, No. 7 is for sure not an appropriate reference for principles of vitamin D signaling.
  3. 3. The ubiquitous expression of VDR should be discussed more differentially, since there are large differences between the tissue. Please refer to respective databases.
  4. 4. A major problem of this manuscript is that the effects of vitamin D are described very superficial, i.e., in most cases any molecular detail, such as regulated genes, is missing. In contrast, much attention is given on VDR polymorphisms providing the misleading impression that they play a major role in the described topic.
  5. 5. In part the paragraphs are very long. Please split them into smaller blocks of information and give your arguments a better structure.

Minor comments:

  1. Vitamin D2 is primarily produced by fungi, Fig. 1 shows this correctly, please adapt the text.
  2. 2. All gene name abbreviations should be in Itallc

Author Response

We sincerely thank the reviewer for its suggestions that offer us the opportunity to improve the quality of our manuscript.

Major comments: 

  1. Some immune cells also express CYP27B1 and produce 1,25D in a para- and autocrine fashion. this should be discussed and illustrated in Fig. 1.

The possible para- and autocrine function for vitamin D signaling in some tissues has been better specified and Fig.1 has been modified accordingly.

  1. Please carefully check the relevance of the chosen references. For example, No. 7 is for sure not an appropriate reference for principles of vitamin D signaling.

References have been carefully checked and updated. 12 more citations have been added.

  1.  The ubiquitous expression of VDR should be discussed more differentially, since there are large differences between the tissue. Please refer to respective databases.

Thank you for this important observation. Differences in VDR tissue distribution have been added to the text; lines 98-104

  1. A major problem of this manuscript is that the effects of vitamin D are described very superficial, i.e., in most cases any molecular detail, such as regulated genes, is missing. In contrast, much attention is given on VDR polymorphisms providing the misleading impression that they play a major role in the described topic.

We thank the reviewer for this interesting comment. We tried to balance the focus on the genetic regulation in vitamin D by adding a paragraph at the end of the “Vitamin D and VDR expression and biologic effects” section, and we shortened the part about the VDR polymorphisms. Since the potential regulated genes are numerous and their functions are often not still completely clear, we tried to focus more on the functional effect of vitamin D/VDR pathway on gut-joint axis than on specific regulated genes.

  1. In part the paragraphs are very long. Please split them into smaller blocks of information and give your arguments a better structure.

The manuscript sections have been rearranged in accordance with your suggestions.

Minor comments:

  1. Vitamin D2 is primarily produced by fungi, Fig. 1 shows this correctly, please adapt the text.

The text has been modified as suggested

  1. All gene name abbreviations should be in Italic

Gene names have been corrected

Round 2

Reviewer 2 Report

none